# Temperature Changes and Their Impact on Drought Conditions in Winter and Spring in the Vistula Basin

**Emilia Karamuz and Renata J. Romanowicz *** 

Institute of Geophysics Polish Academy of Sciences, 01-452 Warsaw, Poland; emilia_karamuz@igf.edu.pl
* Correspondence: romanowicz@igf.edu.pl

**Abstract:** Inter-annual variability of hydro-meteorological variables indirectly influence soil moisture conditions in winter and early spring seasons. The interactions between temperature changes and drought conditions are studied by an application of statistical analyses of minimum temperature (Tmin), consecutive days with temperature exceeding the 0 °C threshold value, the number of melting pulses in the winter season and Standardized Evaporation Precipitation Index (SPEI). Additionally, shifts in the onset of days with spring temperature and snow cover occurrence are analysed. A Mann–Kendall test is applied for the trend analysis. Studies have shown significant changes in thermal characteristics in the winter season over the past 70 years, which affect the moisture conditions in the Vistula River Basin. As a result of those changes, the Vistula Basin is more prone to droughts.

**Keywords:** soil moisture; spring drought; Vistula Basin; statistical analysis; SPEI

## 1. Introduction

Drought presents a complicated research problem. They are the result of the seasonal interplay of precipitation, soil moisture, and snow processes. The propagation from meteorological to hydrological drought strictly depends on precedent moisture conditions. Due to climate change, thermal conditions in the colder half of the year change rapidly, twice as quickly as in the summer season [1]. In Poland, in recent years, dryer and milder winters have been observed [2]. It is disadvantageous for agriculture as it causes a decrease of soil moisture in spring, especially when this is accompanied by the lack of rainfall. If the unfavourable conditions associated with a meteorological drought are prolonged, there is a high probability of a severe hydrological drought in the summer. Dry springs, as Ionita et al. [3] show, have been happening more frequently in recent years due to changes in the appearance of synoptic-scale disturbances in form of atmospheric blocking. In the period 2007–2020, a significant increase in the frequency of occurrence of blocking conditions in transient circulation disturbances over central Europe is observed [3]. As the authors pointed out the years with the highest anomalies in the blocking patterns (e.g., 2007, 2015, 2018, 2019) correspond to the years with significantly increased precipitation deficit. In analyses of temperature changes based on E-OBS gridded data [4], April stands out in the whole of Europe with a strong increase in evapotranspiration rates, with statistically significant changes concentrated in central and western Europe [3]. Analyses also showed very large precipitation deficits over the last 14 year period 2007–2020 [3]. Poland, together with Germany, the Czech Republic, Hungary and Ukraine, is among the countries most affected by the lack of spring precipitation. In the same period (2007–2020), April temperatures of the central and southern Europe were up to 3 °C higher than normal.

Typically, winter in Poland is the time when water resources partially recover. Retained precipitation in the snow cover is an important source of supply in early spring when plants need a lot of moisture to develop. In recent years, we have observed warmer winters in Poland compared to long-term averages [2]. This in turn is reflected in the occurrence of less abundant snow cover. Low-snow winters with a predominance of precipitation in the form of rain have become more frequent [5]. This also leads to a reduction in winter

retention and creates favourable conditions for the development of agricultural droughts in the summer due to the fact that snow cover is linked with the soil water storage [6]. Markonis et al. [7] reported that rain-to-snow season droughts are in decline. There is also noticeable transition from rain-to-snow season droughts to warm-season droughts in regions with a snow cover due to earlier snow melt [8]. The Copernicus Climate Change Service/ECMWF (European Centre for Medium-Range Weather Forecasts) reports that the winter of 2020 was the warmest ever observed over Europe [9]. The spring of 2020 in Poland was extremely dry and cumulative water shortages from year to year contributed to dry soil conditions [10]. Snow cover in Poland is characterised by high intra and inter-annual variability [11] with no clear, statistically significant trends in snow cover depth [12,13]. Some studies [13–15] report a slight decreasing trend of snow cover characteristics (number of days with snow cover, depth of snow cover, occurrence frequency) in Poland during the second half of the 20th century, but no changes were found for a longer period. According to IPCC report [1] the Northern Hemisphere spring snow cover has decreased in extent.

At the beginning of the 21st century, Europe was hit by a series of severe, long-lasting summer heatwaves and droughts (i.e., 2003, 2010, 2013, 2015, 2018, 2019) [2,3,10]. These events were also observed in Poland. The importance of winter retention in Poland has been exemplified by the very dry soil moisture conditions in the catchment after the severe 2018–2019 drought. The meteorological situation in autumn 2019 and winter 2019/2020 brought precipitation close to the multi-annual norm. Despite this, hydrological conditions still remained unfavourable [2]. The vast majority of winter precipitation was in the form of intensive rain, which translated into very high surface runoff. The moisture conditions were also adversely affected by extremely high temperatures in February 2020 [16]. This resulted in a very dry spring 2020 and a development of extreme drought. A sad but very meaningful example of how bad the moisture conditions were at that time was the huge wildfire in the Biebrza National Park (one of the largest and most well-preserved areas of natural swamps and peatlands in the continental biogeographical region) that consumed 10% of the entire park area [17].

In Poland rain-fed agriculture has been prevailing and soil moisture deficits, especially in the spring season, lead to a significant decrease in agricultural productivity. According to Piniewski et al. [18] droughts as severe as those that occurred in Poland in 2015, 2018 and 2019 are projected to be more frequent in the future and to affect larger areas.

Winter and especially spring hydro-climatic conditions are very important for the upcoming summer season. Hänsel et al. [19] have shown that last years have brought a decrease in spring precipitations totals and their significant downward trend is observed over the central part of Europe. The analyses carried out confirm that antecedent moisture deficit is correlated with the number of hot days and maximum heatwave duration, in summer, both at European [20] and global scales [21]. Quesada et al. [22] pointed out that spring shortage of soil moisture, in combination with different weather types, are important factors influencing the occurrence of hot days in summer over Europe. Hanel et al. [23] in their study noted that the recent droughts over Europe often tended to develop during the vegetation period, whereas the previous droughts over the last 250 years were predominantly initiated during the late summer/early autumn. Ionita et al. [3] states that in general, not enough attention has been given to date to the variability of the European climate and its driving factors in the transition seasons (i.e., spring and autumn), although climate-related anomalies and consequent land surface conditions (e.g., soil moisture and/or soil moisture memory, snow cover) during the transition seasons are as important as those in winter and summer.

In the present study, an emphasis is placed on the analysis of thermal conditions in the colder half of the year, which generate changes in the retention processes but also lead to shifts in the thermal seasons and associated phenological phases. Hydro-meteorological characteristics were used as indicators of soil moisture conditions, especially water retention capacity, during snow melt periods. Changing patterns of melting pulses and temporal shifts in the start dates of growing season are very important from the point

of view of surface runoff delay and soil moisture retention at a crucial moment for a plant growth. The reduction of water retention in winter and decreased early spring supply from accumulated snow cover strongly interact with thermic conditions of the summer season. A number of studies indicate that extreme meteorological conditions, including drought and hot extremes, become more frequent. In Poland, in the last decade, there is an evidence of drought signals becoming more frequent and wide-spread [24–26].

There is also a lack of in-depth analysis of hydro-meteorological variables of the transitional period, late winter, early and mid-spring. Motivated by this, we study changes in selected hydro-meteorological indices, from which it is possible to indirectly infer about soil moisture conditions during the transition period looking at changes in the individual months of the colder half of the year. The main aim of this study is to analyse inter-annual variability of hydro-meteorological indices that may indirectly characterize soil moisture conditions in winter and early spring season. In Section 2 we present the methods applied, hydro-meteorological indices, and the study area. Section 3 illustrates the results with a focus on spatiotemporal characteristics of selected indices. In Section 4, the results are discussed in the context of the other studies on that subject. The main conclusions on the inter-annual variability of hydro-meteorological indices are presented in Section 5.

## 2. Materials, Methods and Study Area

The analysis was performed using daily temperature (including daily mean, daily minimum and daily maximum temperature) and precipitation data (both as snowfall and rainfall) from 21 meteorological stations located in the Vistula River Basin (Figure 1) for the period 1951–2020. Data were provided by the Institute of Meteorology and Water Management, State Research Institute (IMGW-PIB) which is responsible for the network of hydro-meteorological measurements in Poland.

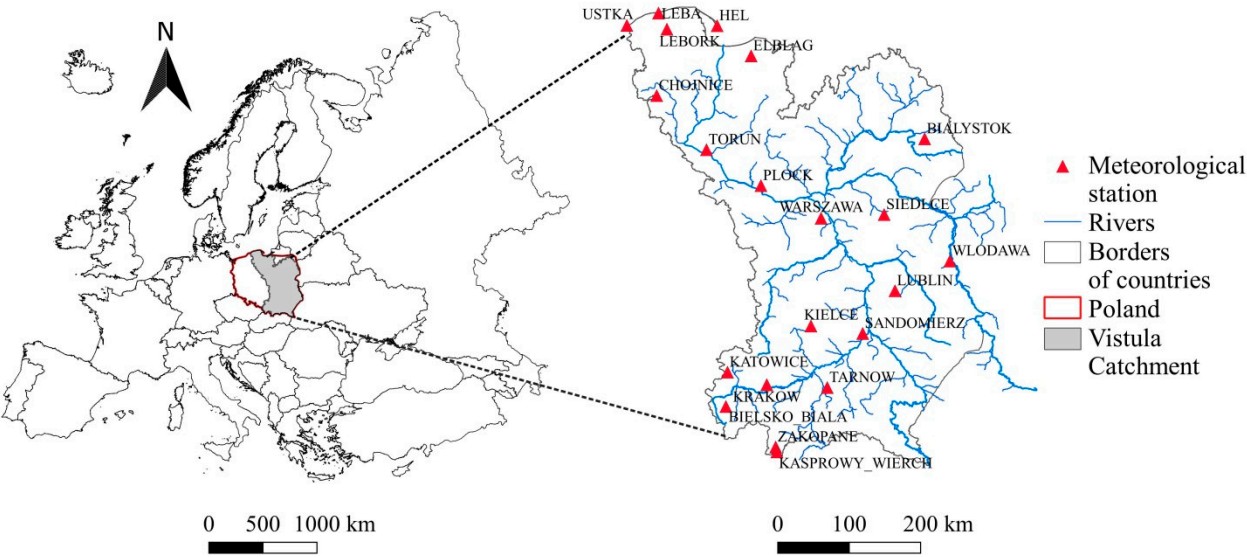

**Figure 1.** Study area with location of meteorological stations.

Hydro-meteorological characteristics indirectly characterizing soil moisture conditions in winter and early spring season were applied.

The analysed indicators included:

- maximum minimum temperature—$maxT_{min}$,
- number of days with $T_{mean} > 0\ °C$—$NDT_{mean}$,
- number of days with $T_{min} > 0\ °C$—$NDT_{min}$,
- maximum length of consecutive days with $T_{mean} > 0\ °C$—$maxCDT_{mean}$,
- maximum length of consecutive days with $T_{min} > 0\ °C$—$maxCDT_{min}$,

- number of melting pulses based on differences between daily maximum and minimum temperature—$NMP_{max-min}$,
- number of melting pulses based on a day-to-day differences in daily mean temperature $NMP_{mean}$,
- dates of the first days with spring temperatures,
- maximum snow cover depth—$maxSCD$,
- number of days with snow cover—$ND\_SC$,
- Standardized Evaporation Precipitation Index—$SPEI$

Threshold level (TL) method was applied to identify consecutive days without precipitation and winter season days with temperature exceeding the 0 °C. The tendencies of changes of selected indices were estimated using trend analysis by Mann–Kendall method [27,28]. This method is one of the most popular techniques to estimate trends in time series because in the Mann–Kendall test there are no assumptions related to the distribution of residuals compared with linear regression [29]. The Mann–Kendall test for trend is based on a rank correlation test for the observed values and their order in time. Test statistics (S) is calculated from the following equation:

$$\mathrm{S} = \sum_{k=1}^{n-1} \sum_{j=k+1}^{n} sgn(x_j - x_k) \tag{1}$$

where:

$$sgn(x_j - x_k) = \begin{cases} +1 & if \ (x_j - x_k) \ > 0 \\ 0 & if \ (x_j - x_k) \ = 0 \\ -1 & if \ (x_j - x_k) \ < 0 \end{cases} \tag{2}$$

The significance of a trend is tested by comparing the standardized test statistics $Z$ with the standard normal cumulative distribution at a selected significance level.

$$Z = \begin{cases} \frac{S-1}{\sqrt{var(S)}} & if \ S > 0 \\ 0 & if \ S = 0 \\ \frac{S+1}{\sqrt{var(S)}} & if \ S < 0 \end{cases} \tag{3}$$

Positive values of $Z$ statistics indicate an increasing trend, while negative $Z$ values indicate a decreasing trend. The trend is statistically significant at $\alpha$ level 0.05 when the absolute value of $Z$ is higher than 1.96.

The Standardized Precipitation-Evapotranspiration Index (SPEI) [30] was used to analyse inter-annual variability of the wetness conditions in the studied catchment. Determination of SPEI based on the statistical analysis of atmospheric water balance, calculated as a difference between precipitation and potential evapotranspiration. Potential evapotranspiration was obtained using the temperature based Hamon method [31,32]. A log-logistic probability distribution was used to fit the empirical distribution of sums of differences between precipitation and potential evapotranspiration aggregated over a chosen time period. The quantiles of estimated log-logistic distribution were transformed into standard normal variables. In this work, the one and three month aggregation period was used. From the obtained SPEI1 and SPEI3 series, those corresponding to the individual winter (December, January, February) and spring (March, April, May) months of the studied years were extracted. This allowed to trace the tendencies of wetness changes in the studied period of analysed seasons. The SPEI values below -1 indicate drought conditions.

All calculations were performed using MATLAB software. The maps were prepared in the free software QGIS.

Characteristics of hydro-meteorological conditions refer to the catchment area of the Vistula River (169,000 km$^2$), the largest river in Poland. The catchment area occupies 54% of the total area of Poland. The land use structure is dominated by arable land (66%). Forests and semi-natural ecosystems cover 29% of the catchment area. Approximately 3% of the

area is urbanised. Based on the Köppen–Geiger climate classification (adapted by Peel et al. [33]), the Polish climate is classified as cold, without dry season and with a warm summer (Dfb). In the Vistula catchment (as in the whole of Poland), climate type Dfb dominates. In the south, there is also type Dfc (cold, no dry season, cold summer) and in the highest mountain parts type—ET (polar, tundra). A detailed description of the study area is given in Karamuz et al. [34].

## 3. Results

### 3.1. Inter-Annual Variations of the Maximum Minimum Daily Temperature

From the point of view of characteristics of potential conditions for accumulation of snow cover and redistribution of soil moisture, the minimum daily temperature is a limiting determinant of the condition of the active layer of soil and its readiness to interact with potential precipitation. If the temperature is below zero, there are conditions for snow accumulation; if it is above zero, even if the precipitation is in the form of snow, it will quickly melt away. Too early high minimum temperatures postpone the moment of moisture supply from the thawing (if part of the precipitation has been accumulated in the form of snow), thus missing the moment of greatest demand by plants in the initial phase of their growth. The process of thaw itself (its rapidity) will depend on temperature passing over the 0 °C threshold value (analysis of number of melting pulses—see Section 3.3 in this Chapter). The extremely warm winters and springs observed in recent years in Poland in the 2018 and 2020 [10,35] point out the need for verification of the long-term directions of change which seems to be very important from the perspective of potential changes in the accumulation of snow and the timing of thawing.

Changes in the maxima of annual minimum temperatures for all examined stations are presented in Figure 2. The degree of change per decade is expressed by the size of a marker, and the significance of changes by the coloured scale. Statistically significant changes are marked with a darker colour scale (dark red colour). The directions of changes for all stations are consistent. At all analysed stations, an increase of daily minimum temperature maxima over 70 years is observed. For December, the range of change varies from 0.2 °C to 0.4 °C, they are not as pronounced as in the following month. For January the changes are larger, up to 0.6 °C per decade (the most substantial changes at the stations Warszawa, Siedlce and Włodawa). For both months the largest changes are observed in the central part of the catchment. In the case of January, for most of the stations, positive directions of changes are statistically significant at a 0.05 level. For December, statistically insignificant changes prevail. In February, only in the south-western part of the study area, positive directions of changes are not statistically significant (Figure 2). Changes in that month vary from 0.2 °C to 0.52 °C. March, similarly to January, is characterized by statistically significant changes for the entire area under study. It is worth noting that in that month the positive directions of changes are very clearly visible (compared to January) also at the stations in north-western Poland and the changes range from 0.2 °C to 0.58 °C. In April, stations with statistically significant changes continue to cluster in the north-western part of the study area, in the central belt and the south-western part. In that month we still observe the predominance of stations with statistically significant changes (increase within the range of 0 °C–0.56 °C). In the following month, there are only three stations with the statistically insignificant increase of maximum minimum temperature. In the case of May, we observe the largest change per decade at the Hel station (0.70 °C). At five stations, the changes are at the level of 0.46 °C–0.56 °C.

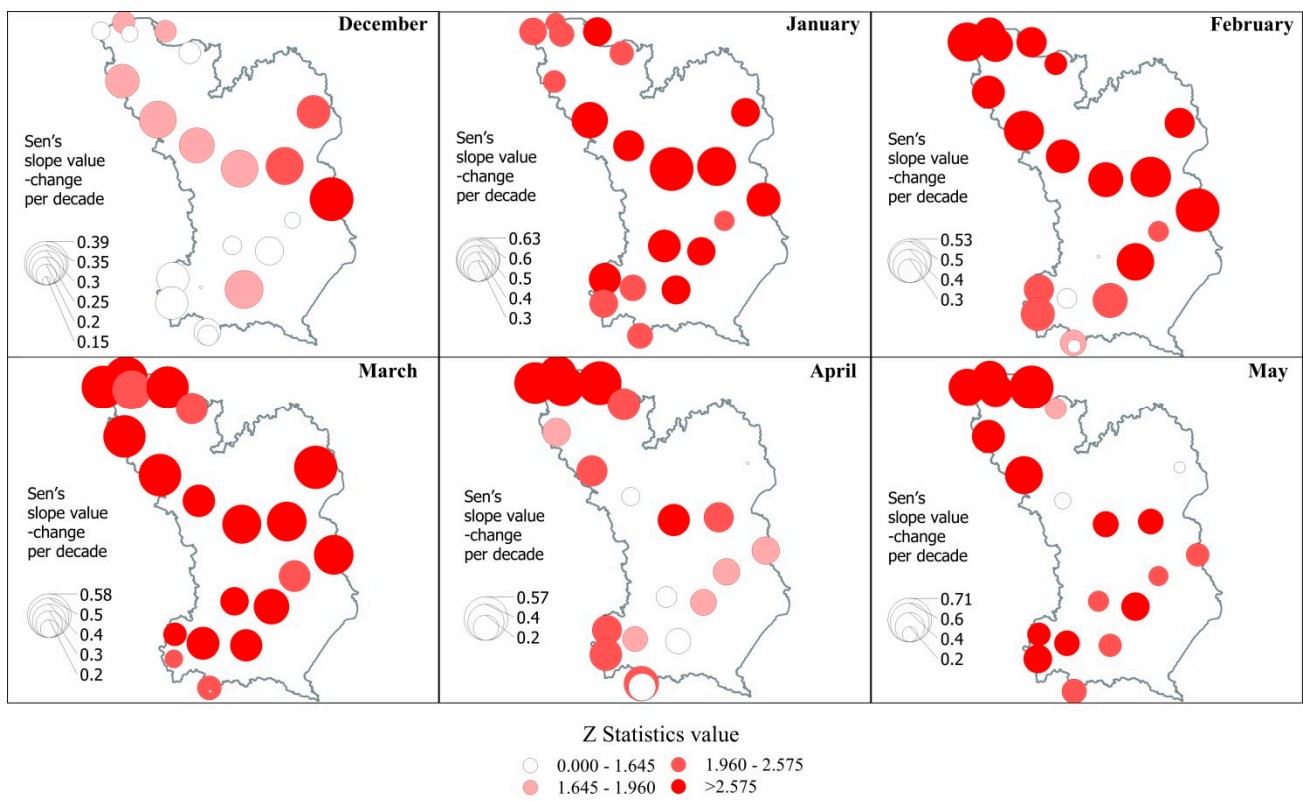

**Figure 2.** Sen's slope values (change per decade) and significance test results for changes in the maximum minimum temperature for the Vistula catchment for the period 1951–2019 for December, January, February (maps in the **upper** row), March, April and May (maps in the **bottom** row).

### 3.2. Variability of the Length of Periods with Positive Temperatures in the Winter Months

The number of days and length of periods with positive temperatures in winter months is the other indicator describing unfavourable conditions for snow accumulation and showing also potential changes in the transition from rain-to-snow season drought to warm-season drought [8].

Figure 3 presents changes in the number of days with average daily and minimum daily temperatures above 0 °C for all examined stations. Similarly, to the previous indicator, a degree of change per decade is expressed by the size of the marker, and the significance of the changes by the coloured scale. Statistically significant changes are marked with a darker colour scale (dark red colour denotes the most significant changes).

Both in the case of daily average temperature and daily minimum temperature an increase in the number of days with positive temperature in the winter season is observed (Figure 3). At all stations except mountainous station, Kasprowy Wierch these changes are statistically significant. The largest changes per decade (3–3.8 days for mean daily temperature and 4–4.7 days for minimum daily temperature) can be observed in the north-western part of the catchment. Quite large changes are also noted in the central belt, for the Toruń, Płock, Warszawa, Siedlce and Włodawa stations.

At the examined stations, we observe also an extension of periods with successive days with positive temperature (Figure 4). This tendency indirectly translates into the extension of disadvantageous conditions for accumulation and preservation of snow cover resulting in the deterioration of soil moisture conditions at the beginning of the spring. In the case of mean daily temperature (left hand panel in Figure 4) at most stations, those changes are statistically significant. If we look at the results for the minimum temperature, changes in the south-western part of the catchment are not statistically significant (yellow colour in shades of grey).

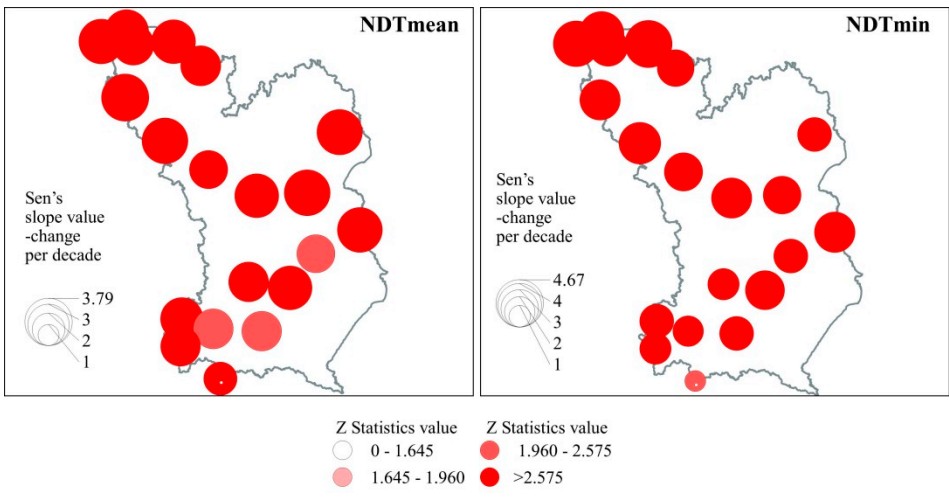

**Figure 3.** Sen's slope values (change per decade) and significance test results for changes in the number of days with average daily (**left** map) and minimum daily (**right** map) temperatures above 0 °C for the Vistula catchment (considered data period—1951–2020).

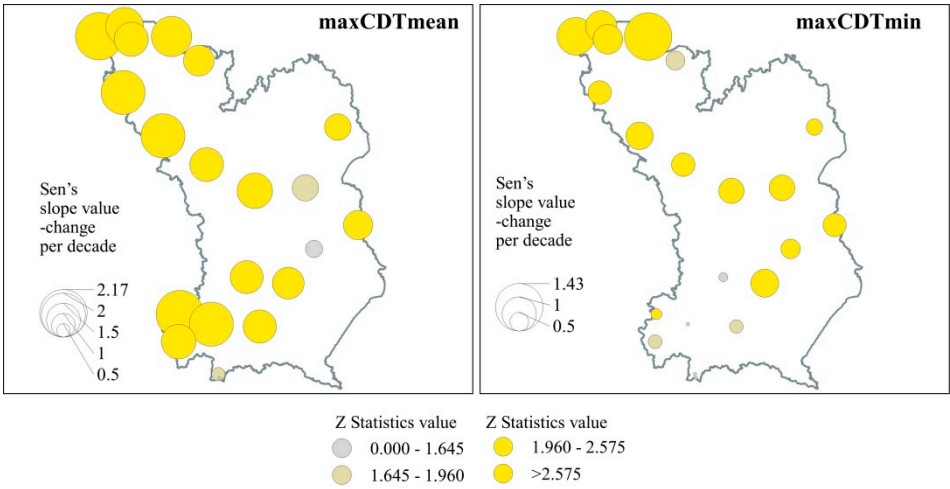

**Figure 4.** Sen's slope values (change per decade) and significance test results for changes in the maximum length of consecutive days with mean (**left** map) and minimum (**right** map) temperature above 0 °C for the Vistula catchment (considered data period—1951–2020).

*3.3. Changes in the Number of Melting Pulses*

In the study, melting pulses were defined as an event when during one day (differences between maximum and maximum temperatures) or between consecutive days (differences between daily averages) the temperature crossed a threshold temperature value (0 °C). The number of melting pulses in a given winter season approximates the dynamics of snow accumulation and melting conditions. This approximation is not very specific, however, it gives information about the basic direction of changes in daily temperature regime in winter and can show potential moisture conditions in an early spring season.

In a contrary to previous indicators, no clear trends are apparent regarding changes in the number of melting pulses. For the number of melting pulses based on differences between daily maximum and minimum temperature ($NMP_{max-min}$), changes indicating a decrease in the number of melting pulses prevail. These decreasing trends are statistically significant at six locations (Figure 5—left hand panel). Changes indicating an increase in the number of melting pulses have a smaller range and are statistically insignificant. Only for mountainous station Kasprowy Wierch we observe slight increase of melting pulses that are statistically significant.

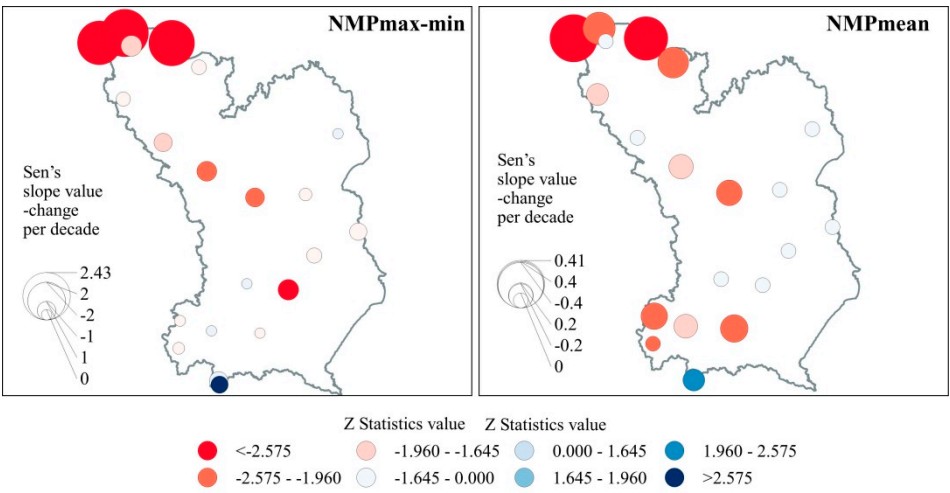

**Figure 5.** Sen's slope values (change per decade) and significance test results for changes in the number of melting pulses based on differences between daily maximum and minimum temperature (**left** map) and based on day-to-day differences in daily mean temperature (**right** map) for the Vistula catchment (considered data period—1951–2020).

When it comes to the number of melting pulses ($NMP_{mean}$) based on day-to-day differences in daily mean temperature (Figure 5—right hand panel) we observe statistically significant negative trends in the south-western part of the catchment (three stations) and at most of the stations located in the north-west part of the catchment (stations located at the Baltic Sea). In the other parts, the changes are not statistically significant.

### 3.4. Shifts in the Onset of Days with Spring Temperature

The disadvantageous direction of changes is also confirmed for the arrival of days with spring temperature. In this study, two indices were analysed to determine the onset of the first periods with spring temperatures: a minimum of five consecutive days with temperature in the range 5 °C–15 °C (selection of that temperature interval follows Romer's methodology [36] and a maximum period with consecutive days with temperature in the same range (days in the colder half of the year were considered).

In the study area, the first five consecutive days with spring temperatures appear on average around 92 (88) day of the year. Values in parentheses refer to stations excluding mountainous areas. The earliest such days occur on an average around 14 (6) day of the year and the latest around a day 127 (123) of the year. Average dates of appearance of days with spring temperature at individual stations are presented by arrows (in Figure 6), whit an angle from the north direction informing about an average date (from 1951–2020 period) of the occurrence of these days. When it comes to the maximum length of periods with temperature in the range of 5 °C–15 °C, they appear in the study area on average around 121 (118) day of the year. The earliest they can be expected around 75 (71) day of the year and the latest around 186 (184) day of the year. In Figure 7 average dates of occurrence of such days at individual stations from the period 1951–2020 are also presented by arrows.

At all stations, we observe an arrival of first five consecutive days with spring temperatures on average 2.4 days earlier (Figure 6—left hand panel). For most stations, these changes are statistically significant (Figure 6—right hand panel). The largest changes (dark green colour of arrows) are observed in the north-western part of the catchment area, reaching 3.57 days at Łeba. They are also quite high in the southern part, in Katowice, Kraków and Tarnów (statistically significant only in Kraków). In the middle zone of the area under study, we observe statistically significant changes within two to three days.

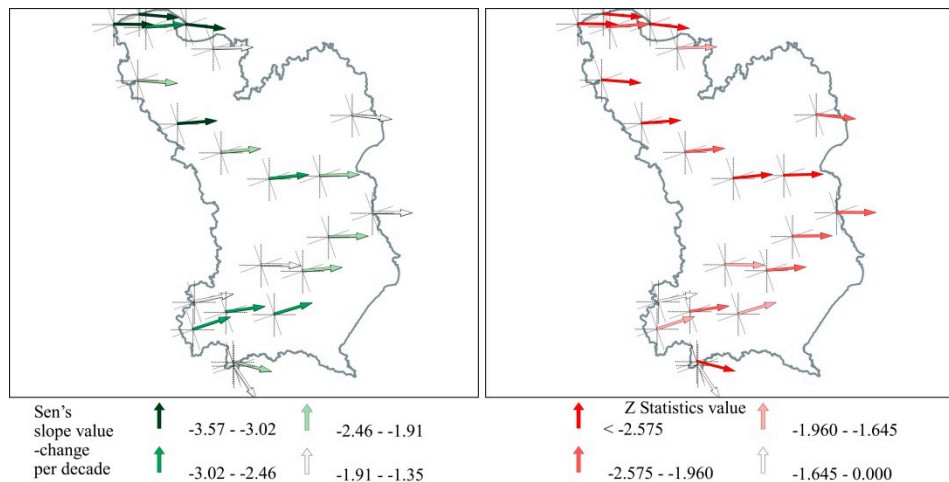

**Figure 6.** Sen's slope values (**left** map) and significance test results (**right** map) for changes in the onset of the first days with spring temperatures (minimum five consecutive days with average daily temperature between 5 °C and 15 °C) for the Vistula catchment (considered data period—1951–2020).

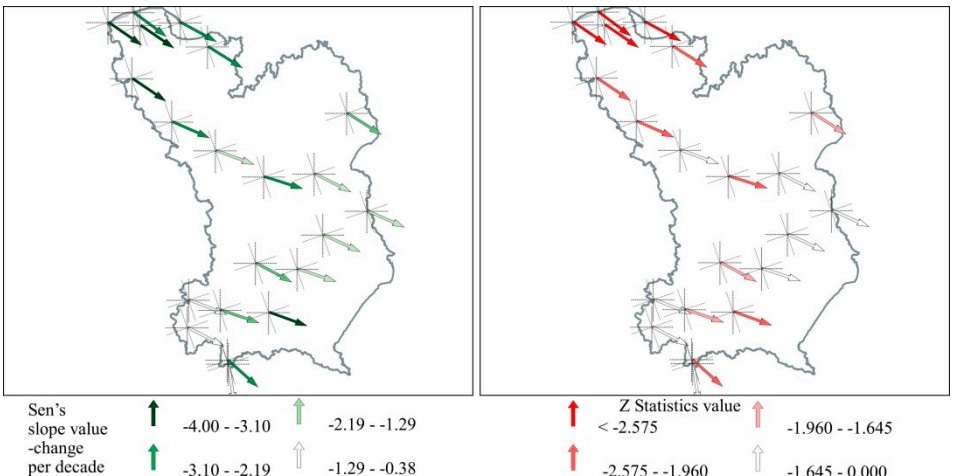

**Figure 7.** Sen's slope values (**left** map) and significance test results (**right** map) for changes in the onset of days with spring temperatures (the longest period of consecutive days with average daily temperature between 5 °C and 15 °C) for the Vistula catchment (considered data period—1951–2020).

In the case of maximum number of consecutive days with temperatures in the range 5 °C–15 °C we observed the same character of changes, with the difference that there are fewer stations where the changes are statistically significant. On average, the maximum periods with spring temperature appear earlier by 2.3 days (Figure 7—left hand panel). The largest changes (dark green colour) occur in Lębork (four days earlier). A distinct (statistically significant) shift in the occurrence of days with spring temperature occurs in the northern part of the studied catchment.

### 3.5. Inter-Annual Variations of the Monthly Sum of Precipitation

The average precipitation for the examined period for the whole study area for December through May was 46.3 mm (40.7 mm excluding mountain stations). For the winter period after excluding the mountain stations the mean precipitation was 36.5 mm (40.9 mm with mountain stations). In the spring months, these values were higher and amounted to 44.9 mm (without mountain stations) and 51, 6 mm (with mountain stations) respectively. If we look at individual stations, the average precipitation for winter months ranged from 27.2 mm in Sandomierz to 47.4 mm in Lebork (115.9 mm on Kasprowy Wierch—

mountainous station). The average precipitation in the spring months varied from 34.9 mm on Hel to 81.2 mm in Bielsko Biała. In wintertime at most of the stations, the lowest average precipitation occurred in February (in the case of two stations Zakopane and Bielsko Biała the lowest average precipitation fell in January). In turn, the highest average precipitation in the winter months at all stations occurred in December. In the spring period, the lowest average precipitation fell in March and the highest in May.

As can be seen in Figure 8, the precipitation changes can be positive or negative, unlike those of maxima of annual minimum temperatures. In winter (Figure 8—upper panels) changes per decade at individual stations ranged from −30 mm to 41 mm. The highest range of changes was observed for February (−30 mm–41 mm), the lowest for December (−17 mm–19 mm). In spring (Figure 8—lower panels), the range of changes is larger (from −28 mm to 90 mm). The largest differences in directions of changes are observed in May (−17.3 mm–90.0 mm) and the smallest in March (−11.7 mm–32.4 mm).

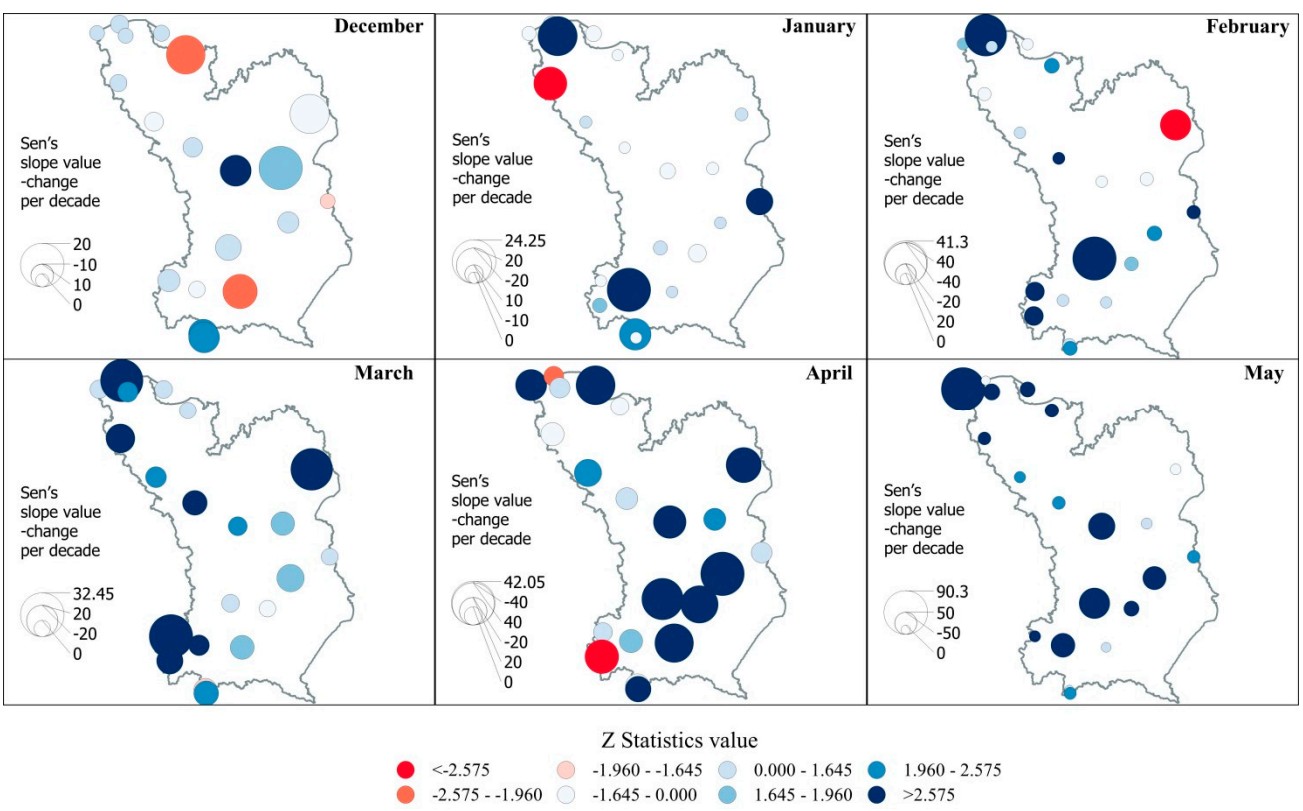

**Figure 8.** Sen's slope values (change per decade) and significance test results for changes in monthly precipitation totals for individual months for the Vistula catchment (considered data period—1951–2020).

In winter months changes indicating a decrease of monthly sums of precipitation were observed at a small number of stations (five stations in February, six stations in December and nine stations in January), whereas statistically significant changes (marked with a darker red colour) were noted only in few cases (two stations in December and one station in January and February). An increase in monthly sums of precipitation was observed at most of the stations (11 stations in January, 15 stations in December and 16 stations in February). The largest number of statistically significant changes (dark blue colour) was in February (eight stations with 6.5 mm—41.26 mm). In this month, the largest increase over 10 years was recorded in Kielce (41.26 mm) and Leba (39.8 mm). In December statistically significant increases in precipitation appeared at five stations and in January at four stations. The range of increase was 13.6–18.2 mm (the highest in Bielsko Biała) and 15.4–24.2 mm (the highest in Karaków) respectively.

In spring a decrease in monthly sums of precipitation was observed at two stations in March and May and at five stations in April. In May, at none of the stations, this direction of change was statistically significant. In March, only at one station, the decrease in precipitation was statistically significant, and in April at two stations. Similarly to winter, in spring an increase in monthly sums of precipitation was observed at most of the stations (20 stations in March, 19 stations in May and 16 stations in April). The largest number of statistically significant changes was in May (16 stations). The range of increase was 19.2 mm—90.3 mm (the highest in Ustka, quite high increases were also observed in Kielce—67.0 mm, Warszawa—59.4 mm, Krakow—53.9 mm and Lublin—52.6 mm). Both in March and in April statistically significant changes were observed at 11 stations. The range of increase was 6.6–32.4 mm (the highest in Katowice) and 12.4–42.0 mm (the highest in Lublin) respectively.

Long-term trends of changes in precipitation amounts in individual months (I–IV) did not show any decrease in the studied catchment over the years 1951–2020. However, there are winters and springs with precipitation that differ significantly from the multi-year norm (Figure 9). In studied catchment, seasonal precipitation during colder half of the year should improve the soil moisture conditions after warm or warm and dry spring and summer. Otherwise, at the beginning of the next warm season, moisture conditions can be too low and can lead to a drought. Winter–spring supply is very important, negative deviations from annual norms are very unfavourable especially in the initial phase of the growing season. In the years when Poland and Europe experienced severe droughts (2003, 2015, 2018, 2019, 2020), most of the study area had negative precipitation anomalies of various magnitudes, varying spatially (Figure 9). Attention is drawn by very unfavourable conditions in April in the following three years (2018, 2019, 2020), whose apogee was observed in 2020. Cumulative water shortages in recent years have contributed to the development of a severe drought in 2020.

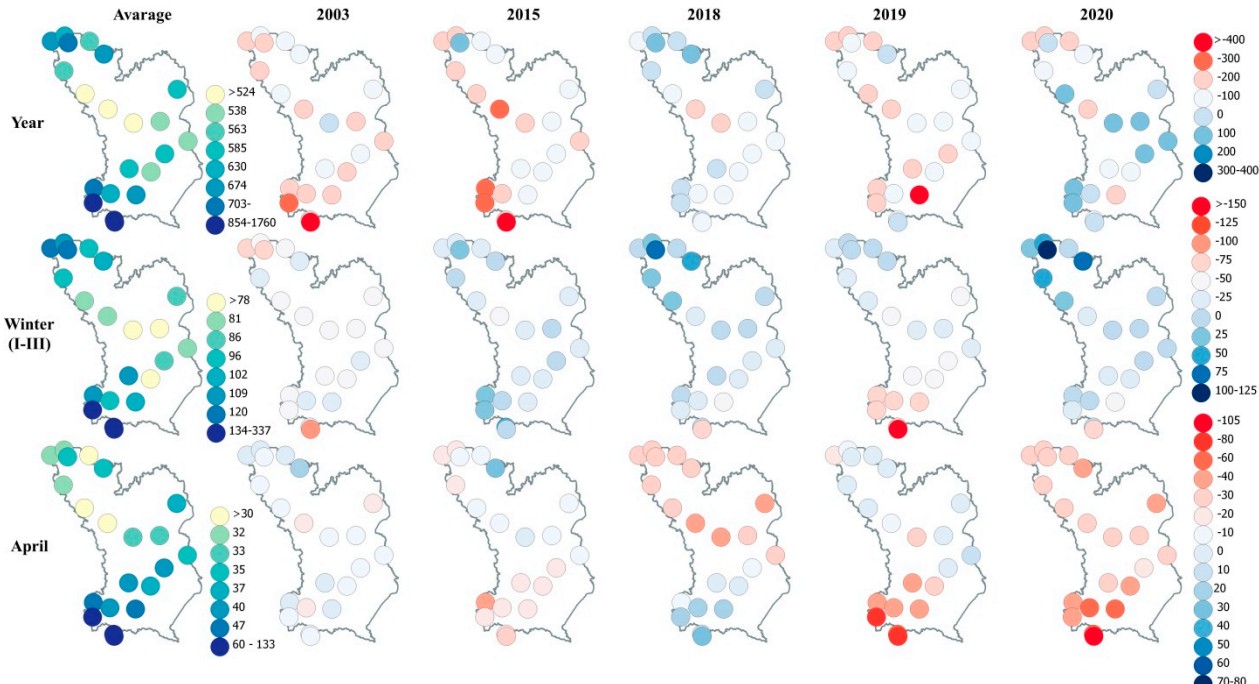

**Figure 9.** Precipitation anomalies in the years 2003, 2015, 2018, 2019 and 2020 from the average precipitation for the period 1951–2020. The top row shows anomalies for a mean annual precipitation, the middle for an average winter precipitation and the bottom panel shows anomalies for an average April precipitation.

### 3.6. Changes in the Snow Cover

In the studied area the mean depth of snow cover in January, February and March in 1951–2020 varies from 3.5 cm in Płock to 9.8 in Białystok (except mountainous stations, where the snow cover depth is considerably higher). The average snow cover depth for the whole catchment area (excluding mountain areas) is 5.7 cm. The mean maximum snow cover depth in the examined period ranges from 13.5 cm in Płock to 31.5 cm in Bielsko Biała, with the average for the whole catchment amounted to 20.7 cm (except mountainous stations).

Spatial variability of temporal changes in maximum snow cover depth in winter months (January, February March) in 1951–2020 is presented in Figure 10 (left hand panel). Negative direction of changes was observed at most stations, but only three of them were statistically significant (Zakopane——3.3 cm per decade, Siedlce——1.7 cm and Krakow——1.5 cm). Four stations did not show any changes in snow cover depth (Lębork, Elbląg, Płock and Lublin).

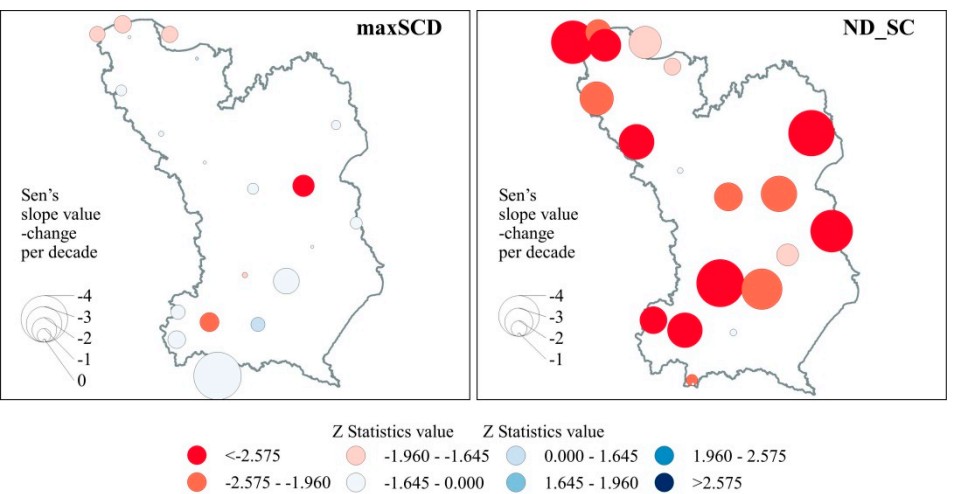

**Figure 10.** Sen's slope values (change per decade) and significance test results for temporal changes in the maximum value of snow cover depth in winter in the selected months, i.e., JFM (**left** map) and number of days with snow cover in the cold half of the year (**right** map) for the Vistula catchment (considered data period—1951–2020).

In the case of a number of days with snow cover, it varies from 36 days (Ustka, Łeba, Lębork, Płock) to 58 days (Białystok, Lublin, Włodawa, Kraków) per year (except mountain stations). At all stations a decrease of a number of days with snow cover is observed, and what is important in majority of them they are statistically significant (14 stations). Negative trend of changes is clearly visible in the entire catchment area (Figure 10—right hand panel). The largest decreases per decade are observed in Ustka (4.2 days), Białystok (4.4 days), Włodawa (4.1 days), Kielce (4.5 days) and Bielsko Biała (3.9 days).

### 3.7. Inter-Annual Variability of Winter and Spring SPEI Values

In many cases soil moisture observations are not available and require expensive measurement campaigns to be obtained. Therefore, in many studies indices characterising drought conditions are used as an approximation of this variable [37,38]. We use the SPEI [30] calculated at a one and three month time scale as a proxy for soil moisture in winter and spring months.

Spatial variability of temporal changes in SPEI1 for particular months in 1951–2020 is presented in Figure 11. Directions of changes are not consistent and in most cases, they are statistically insignificant. In the case of December, January, February and March we observe a similar number of changes indicating improvement and deterioration of wetness conditions in the analysed locations, while a positive trend is observed in the northern

part of the catchment and a negative one in the southern part. In April, negative trends in SPEI1 values were observed at all stations, statistically significant in north-western part of the analysed catchment. In May, there is a clear predominance of changes indicating a tendency to improving moisture conditions over the last 70 years.

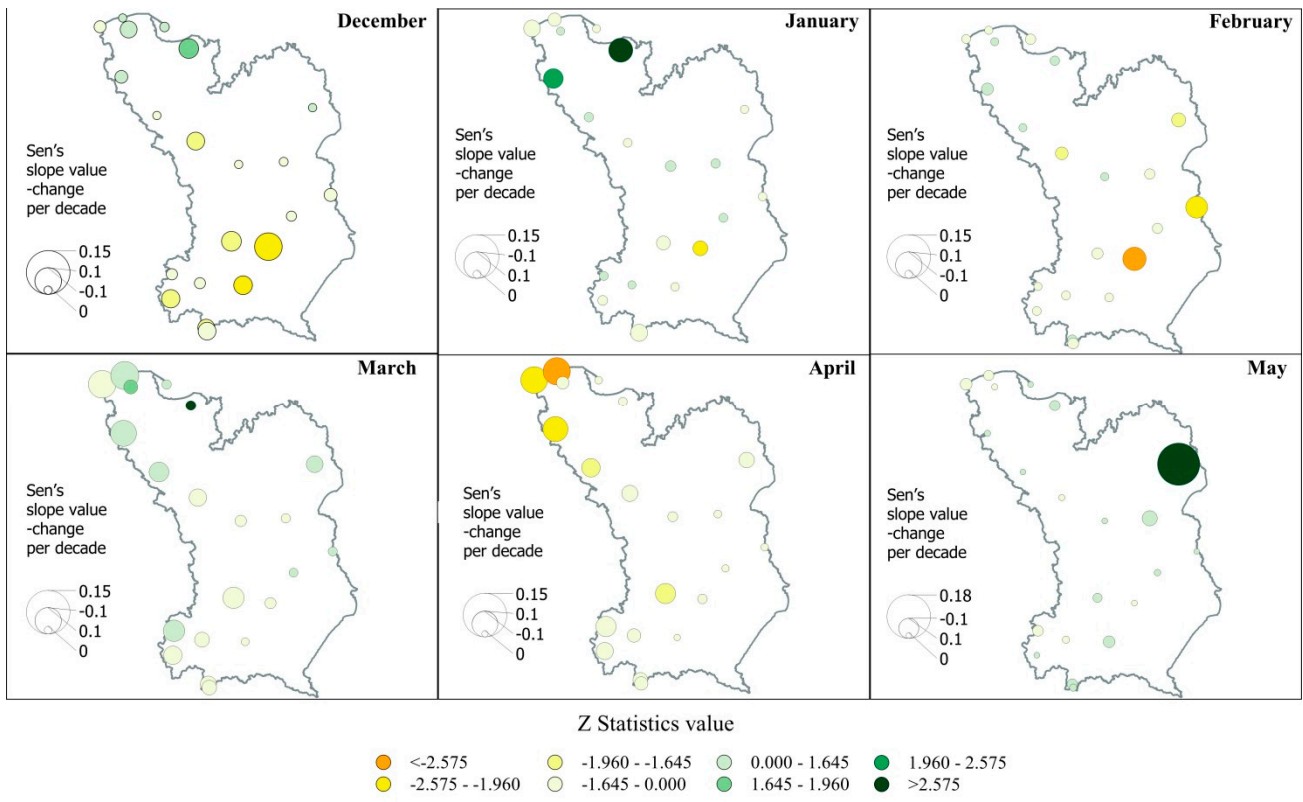

**Figure 11.** Sen's slope values (change per decade) and significance test results for temporal changes in the SPEI1in the selected months, i.e., D, J, F, M, A, M for the Vistula catchment (considered data period—1951–2020).

Spatial variability of temporal changes in SPEI3 for winter and spring in 1951–2020 is presented in Figure 11. In winter, changes indicating deterioration of wetness condition prevail (south part of the catchment). In the spring, changes are very small (statistically insignificant in most cases). Comparison of those results with monthly SPEI1 indicates that we get more insight into drought variability using the later.

## 4. Discussion

From the results obtained, it can be concluded that the directions of changes in the winter and spring periods over the last 70 years indicate a deterioration of conditions that help to maintain a favourable moisture situation at the beginning of the growing season. Based on the obtained results regarding an increase of max-min temperature in the winter–spring period, one can state, that the directions of changes for the whole studied area are unfavourable for soil moisture conditions. The observed statistically significant increase in the maxima of annual minimum temperatures (Figure 2) contributes to the disappearance of winter retention and an increase of the risk of water supply deficiency in the critical moments of the growing season. Moreover, this unfavourable state persists when superimposed by the lack of spring precipitation (such a situation occurred in the 2003, 2015, 2018, 2019 and 2020 as illustrated in Figure 9), resulting in the development of hydrological drought extending into the summer and autumn months.

Prolonged precipitation deficits also translate into an unfavourable hydrological situation and extend the drought period. Additionally, a statistically confirmed lack of winter accumulation (statistically significant increase in the number of days with average

and minimum temperature above zero (Figure 3) in winter, decrease in days with snow cover—Figure 10) as well as a deficit of spring precipitation can make this phenomenon perennial. As stated by Pińskwar et al. [10] even if months occur with higher than the multi-year average sum of precipitation, this excess may not be sufficient to compensate for the accumulated deficit. It often happens that during such months there is one episode (lasting up to 2–4 days) with intensive rainfall translating into a high monthly precipitation sum, which is quite quickly drained from the catchment as surface runoff, improving the soil moisture situation for a very short time. Some indicators describing drought do not cope with such events and may mask a drought event, especially if the analysis is based on a short data series.

The study of Pińskwar et al. [10] showed a decrease in soil moisture in Poland at the depths 0–10 cm, 10–40 cm, 40–100 cm and 100–200 cm over the considered period (2000–2020). As stated by the authors, the shift towards drier soil moisture conditions resulted from the occurrence of dry and warm years as well as high temperatures in all seasons. In addition, our study confirms the existence of unfavourable thermal conditions, observed in the period 1951–2020 distinct tendencies of increasing temperature in winter (which translates into a lack of accumulation of moisture in the snow cover or it thaws too early) and in the transition period may contribute, as has been observed in recent years, to the occurrence of soil drought at the beginning of the growing season (results in Figure 9).

In the case of both indicators that determine the melting pulses, statistically significant changes indicating a decrease in their number coincide with the positive direction of changes of the previously discussed indicators. In those locations (north-western and central part of the catchment in January, February and March), changes in the maxima of annual minimum temperature, number of days and the maximum length of consecutive days with temperature above 0 °C are the most intense.

The obtained results indicate that there are significant thermal changes in the winter season, which do not favour the replenishment of soil moisture after possible deficits occurring in the warm half of the year (increase in the number of days with positive temperature in the winter months, increase in the maxima of minimum temperature, decrease in the number of days with snow cover and decrease of maximum snow cover depth), as well as maintenance of appropriate moisture conditions at the beginning of the vegetation season (here a special role is played by the melting pulses analysed in this study (Figure 5)—a decrease connected with prolonged periods of positive temperature). Our study confirms also a significant shift in start dates of days with spring temperatures (2.4 days per decade).

Looking at the long-term trends of changes (especially indicators based on the thermal criterion), the problem of spring droughts, which has recently attracted wider attention due to a very deep drought in 2020 and media reports, may occur much more frequently in the future. This is currently a pressing problem as Poland is dominated by rain-fed agriculture.

The results of our study are consistent with Ionita et al. [3]. In their analyses, particular attention was paid to the problem of changes in mid-spring moisture and thermal conditions and showed significant changes for April, arguing that they are caused by an increase in the frequency of anticyclonic weather types during this time (and winter and spring in general).

Considering that previous droughts occurring in Europe (2003, 2015) were initiated by thermal-moisture conditions in spring as a combined effect of high temperatures and low precipitation [23], which was also confirmed in subsequent studies [3] and in the current work, it is reasonable to further investigate the hydro-meteorological conditions of the transitional period, as it may prove to be crucial in assessing the deepening of soil drought and the extension of adverse moisture conditions for inter-annual scale (lack of winter moisture feeding).

The results of the analysis of soil moisture variability described by the SPEI1 show that April is the driest month in all the basin with February on the second place, showing the middle part of the basin as the driest. However, the changes are not strongly pronounced.

On the other hand, the SPEI3 does not give consistent results for the whole basin, with the dry middle and wet northern part in winter (Figure 12, left hand panel) and neutral soil moisture conditions in spring (Figure 12, right hand panel). It seems that smaller time step of filtering (e.g., a week or a day) applied for the derivation of that index would help in refining those results.

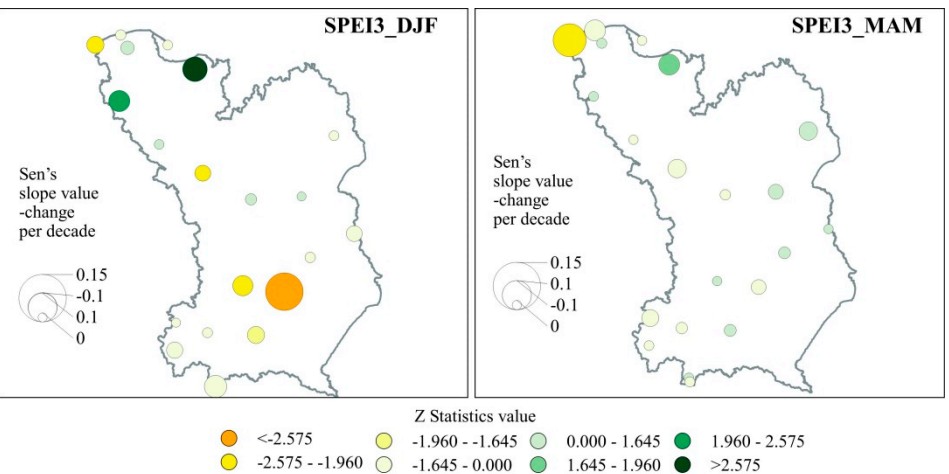

**Figure 12.** Sen's slope values (change per decade) and significance test results for temporal changes in the SPEI3 in the winter (DJM) and spring (MAM) for the Vistula catchment (considered data period—1951–2020).

## 5. Conclusions

We analysed the variability of a number of temperature and precipitation related indices over the last 70-years to show their inter-relation and their influence on soil moisture conditions represented by the SPEI variability. The main focus of this study was placed on two seasons, winter and spring. In the northern hemisphere in Europe, those two seasons set up the antecedent conditions for the summer water balance and the possible drought occurrence. The most important findings of our research are as follows:

1. The directions of changes in the winter and spring periods over the last 70 years indicate a deterioration of conditions that help to maintain a favourable moisture situation at the beginning of the growing season.

2. The research showed a tendency for winter–spring water deficits to cluster in recent years which leads to a worsening of soil moisture conditions and deepening of soil drought.

3. The results show statistically confirmed lack of winter accumulation (statistically significant increase in the number of days with average and minimum temperature above zero in winter).

4. The decrease in days with snow cover as well as a deficit of spring precipitation can make drought problem perennial.

5. Analysis showed significant shifts in start dates of days with spring temperatures (2.4 days earlier per decade).

The results of our research indicate that soil moisture conditions should be monitored on a much larger scale than it is done at present. SPEI based on one month-scale supports our discussion on negative changes towards drier soil moisture conditions occurring in the basin in February and April. The SPEI indicator of soil moisture does not take account of local conditions of soil moisture distribution. The application of a distributed model enabling to get an insight into water distribution within the catchment might help in that respect.

**Author Contributions:** Conceptualization, E.K. and R.J.R.; methodology, E.K.; software, E.K.; validation, E.K. and R.J.R.; formal analysis, E.K. and R.J.R.; investigation, E.K.; resources, R.J.R.; data curation, E.K. and R.J.R.; writing—original draft preparation, E.K.; writing—E.K. and R.J.R.; visual-

ization, E.K.; supervision, R.J.R.; project administration, R.J.R.; funding acquisition, R.J.R. All authors have read and agreed to the published version of the manuscript.

**Funding:** This research received no external funding.

**Institutional Review Board Statement:** Not applicable.

**Informed Consent Statement:** Not applicable.

**Data Availability Statement:** Data supporting reported results can be found in the Institute of Meteorology and Water Management (hydro-meteorological data).

**Acknowledgments:** This work was supported by the project HUMDROUGHT, carried out in the Institute of Geophysics Polish Academy of Sciences, funded by National Science Centre (contract 2018/30/Q/ST10/00654). The hydro-meteorological data were provided by the Institute of Meteorology and Water Management (IMGW), Poland; groundwater data were provided by Polish Geological Institute (PIG).

**Conflicts of Interest:** The authors declare no conflict of interest.

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
