# Peer review of "Temperature Changes and Their Impact on Drought Conditions in Winter and Spring in the Vistula Basin"

_water, doi:10.3390/w13141973_

Round 1

Reviewer 1 Report

Inter-annual variability of hydro-meteorological variables may indirectly influence soil moisture conditions in winter and early spring seasons in the Vistula River basin in Poland. This paper analyzed in detail the inter-annual variability of hydro-meteorological indices, and some meaningful results were obtained. Questions and Suggestions: 1. The title of this paper is the change of temperature and its impact on soil moisture, but this paper mainly analyzes the change characteristics of several indexes. Although these indexes have direct or indirect influence on soil moisture, how does the actual soil moisture change? Relevant research content should be added. If you can't answer the changing characteristics of soil moisture, you need to change the title of the article. 2. The current title does not specify the specific location. 3. In methods, “number of melting pulses based on differences between daily maximum and minimum temperature – NMPmax-min “ and “ number of melting pulses based on a day-to-day differences in daily mean temper- NMPmean differences” are presented, but not used in the text. 4. The analysis method is relatively simple. 5. The discussion should be strengthened, especially with the addition of an analysis of the effect of temperature changes on soil moisture.

Author Response

The authors sincerely thank the reviewers for their comments on the manuscript. We believe that the article, after corrections offers the reader much more clear overview of the investigated problem. In this document, we provide a detailed response to each point raised during the revision. We hope you will kindly appreciate our efforts.

Reviewer 1

Questions and Suggestions:

  1. The title of this paper is the change of temperature and its impact on soil moisture, but this paper mainly analyzes the change characteristics of several indexes. Although these indexes have direct or indirect influence on soil moisture, how does the actual soil moisture change? Relevant research content should be added. If you can't answer the changing characteristics of soil moisture, you need to change the title of the article.

Thank you for your kind advice, much appreciated. According to your suggestions, the title of the article has been changed and, in our opinion, now better reflects the scope of the issues addressed in this study. We agree that the wording soil moisture used in the title suggested that we had data on soil moisture. In order to refer to changes in soil moisture during the studied period, we added the drought index used in many studies to assess soil moisture conditions (which approximates soil moisture). We decided to use Standardized Precipitation-Evapotranspiration Index (SPEI) [Vicente-Serrano et al 2010] (lines 162-174, 430-458).

  1. The current title does not specify the specific location.

Thank you for your comment. The location of the study area was added to the title.

  1. In methods, “number of melting pulses based on differences between daily maximum and minimum temperature – NMPmax-min “ and “ number of melting pulses based on a day-to-day differences in daily mean temper- NMPmean differences” are presented, but not used in the text.

The results of analysis of the mentioned indices are described in paragraph 3.3 - Changes in the number of melting pulses (lines 267 – 292). Abbreviations NMPmax-min and NMPmean have been added to the text in the paragraph 3.3 (line 282 and 288)

  1. The analysis method is relatively simple.

Thank you for your comment. The methods have been expanded. SPEI values were determined for two time scales. A log-logistic probability distribution was used to fit the empirical distribution of SPEI. The quantiles of estimated log-logistic distribution were transformed into standard normal variables (lines 162-174).

  1. The discussion should be strengthened, especially with the addition of an analysis of the effect of temperature changes on soil moisture.

Thank you for this important suggestion. Chapter 4 has been extended. A discussion on the influence of temperature changes on soil moisture has been added (lines 464-472, 495-500, 528- 535). The Conclusions section was also added (lines 537-563).

Reviewer 2 Report

Looking at the title and abstract I expected the paper would evaluate the effect of temperature changes (and number of days above particular thresholds) and snow parameters on soil moisture in winter and spring. But the title is misleading and must be changed, because nowhere in the article the authors specify soil moisture, but only hydro-meteorological characteristics indirectly connected to soil moisture.

The summary of the results should be more specific, the word "may" (L9 and L16) should rather not appear in the abstract  

L37: please explain the abbreviation “E-OBS gridded data”

L66-67: please provide literature for this statement

Fig 1 “Vistula Catchment” is mentioned twice in legend

Please unify the separating sign between integer and decimal numbers in all figures. It should be a dot (.), not comma (,)

Discussion and conclusions chapter: the results and conclusions drawn by the authors should be clearly defined and distinguished from the conclusions drawn from the literature. It is not done and therefore I am not able to judge the scientific value of the article at this stage.

Author Response

The authors sincerely thank the reviewers for their comments on the manuscript. We believe that the article, after corrections offers the reader much more clear overview of the investigated problem. In this document, we provide a detailed response to each point raised during the revision. We hope you will kindly appreciate our efforts.

Reviewer 2

Looking at the title and abstract I expected the paper would evaluate the effect of temperature changes (and number of days above particular thresholds) and snow parameters on soil moisture in winter and spring. But the title is misleading and must be changed, because nowhere in the article the authors specify soil moisture, but only hydro-meteorological characteristics indirectly connected to soil moisture.

Thank you for your kind advice, much appreciated. According to your suggestions, the title of the article has been changed and, in our opinion, now better reflects the scope of the issues addressed in this study. We agree that the wording soil moisture used in the title suggested that we had data on soil moisture. In order to refer to changes in soil moisture during the studied period, we added the drought index used in many studies to quantify soil moisture (which approximates soil moisture). We decided to use Standardized Precipitation-Evapotranspiration Index (SPEI) [Vicente-Serrano et al 2010] (lines 162-174, 430-458).

The summary of the results should be more specific, the word "may" (L9 and L16) should rather not appear in the abstract 

Thank you for your comment. The indicated phrases in the abstract have been changed.

L37: please explain the abbreviation “E-OBS gridded data”

To the best of our knowledge this is not an abbreviation but a name for gridded data generated from observational data gathered by European Climate Assessment and Dataset (ECA&D). For clarity, a citation to the E-OBS data was added in the text (line 37).

L66-67: please provide literature for this statement

Thanks. Done. (now lines 65-66)

Fig 1 “Vistula Catchment” is mentioned twice in legend

Thanks. Corrected

Please unify the separating sign between integer and decimal numbers in all figures. It should be a dot (.), not comma (,)

Thank you for the comment, the separating sign between integer and decimal numbers was unified.

Discussion and conclusions chapter: the results and conclusions drawn by the authors should be clearly defined and distinguished from the conclusions drawn from the literature. It is not done and therefore I am not able to judge the scientific value of the article at this stage.

Thank you for this important advice. Chapter 4 has been changed into Discussion. The Conclusions section was added (lines 537-563).

Round 2

Reviewer 2 Report

Thank you for taking my recommendations into account. Despite the fact that the paper is similar to others in this field, I think it is suitable to be published in Water.